# Self-Assembly of Copper Oxide Interfaced MnO_2_ for Oxygen Evolution Reaction

**DOI:** 10.3390/nano13162329

**Published:** 2023-08-13

**Authors:** Chinna Bathula, Abhishek Meena, Sankar Sekar, Aditya Narayan Singh, Ritesh Soni, Adel El-Marghany, Ramasubba Reddy Palem, Hyun-Seok Kim

**Affiliations:** 1Division of Electronics and Electrical Engineering, Dongguk University–Seoul, Seoul 04620, Republic of Korea; cdbathula@dongguk.edu; 2Division of Physics and Semiconductor Science, Dongguk University-Seoul, Seoul 04620, Republic of Korea; pakar.abhishek@gmail.com; 3Department of Semiconductor Science, Dongguk University-Seoul, Seoul 04620, Republic of Korea; sanssekar@gmail.com; 4Quantum-Functional Semiconductor Research Center, Dongguk University-Seoul, Seoul 04620, Republic of Korea; 5Department of Energy and Materials Engineering, Dongguk University-Seoul, Seoul 04620, Republic of Korea; aditya@dongguk.edu; 6Department of Chemical Engineering, Department of Energy Engineering, Ulsan National Institute of Science and Technology (UNIST), 50 UNIST-gil, Ulsan 44919, Republic of Korea; riteshsoni@unist.ac.kr; 7Department of Chemistry, College of Science, King Saud University, P.O. Box 2455, Riyadh 11451, Saudi Arabia; amarghany@ksu.edu.sa; 8Department of Medical Biotechnology, Dongguk University, 32 Dongguk-ro, Ilsandong-gu, Goyang 10326, Republic of Korea

**Keywords:** CuO/MnO_2_, oxygen evolution reaction, electrocatalyst, stability

## Abstract

Designing efficient electrocatalytic systems through facile synthesis remains a formidable task. To address this issue, this paper presents the design of a combination material comprising two transition metal oxides (copper oxide and manganese oxide (CuO/MnO_2_)), synthesized using a conventional microwave technique to efficiently engage as an active oxygen evolution reaction (OER) catalyst. The structural and morphological properties of the composite were confirmed by the aid of X-ray diffraction (XRD) studies, field emission scanning electron microscopy (FESEM), X-ray photoelectron spectroscopy (XPS), and energy-dispersive spectrometry (EDS). FESEM clearly indicated well-aligned interlacing of CuO with MnO_2_. The OER performance was carried out in 1 M KOH. The assembled CuO/MnO_2_ delivered a benchmark current density (j = 10 mA cm^−2^) at a minimal overpotential (η = 294 mV), while pristine CuO required a high η (316 mV). Additionally, the CuO/MnO_2_ electrocatalyst exhibited stability for more than 15 h. These enhanced electrochemical performances were attributed to the large volume and expanded diameter of the pores, which offer ample surface area for catalytic reactions to boost OER. Furthermore, the rate kinetics of the OER are favored in composite due to low Tafel slope (77 mV/dec) compared to CuO (80 mV/dec).

## 1. Introduction

In the pursuit of achieving a carbon-neutral society and to alleviate the harmful effects of conventional fossil fuels, there is a pressing need to develop sustainable, eco-responsive, and green energy sources. To cater to fossil fuel resource depletion and its associated environmental impacts, in addition to the dynamic geopolitical instabilities of fossil fuel production, energy researchers have explored numerous sustainable energy sources such as wind, biomass, tidal, and solar energy. Among these available energy resources, solar radiation is one of the most abundant natural energy sources, capable of meeting the entire global energy demand, but it suffers from intermittent availability due to geographical variations [1,2]. Furthermore, the sun shines only for a part of the day and its power density varies as the function of geographical altitude imposes another complexity to its wider applicability. Additionally, with the recent progress in photovoltaic devices, the power convergence efficiency hardly reached ~26% [3], further jeopardizing the adequacy of solar power [4]. Among several available renewable energy sources, splitting water (2H_2_O = 2H_2_ + O_2_) to generate useful hydrogen as a fuel and oxygen as a by-product has drawn substantial attraction, as the generated hydrogen can be stored and can be transported at the required site. However, water electrolysis is an energy-intensive process, and its efficiency is limited by the sluggish kinetics of the oxygen evolution reaction (OER) at the anode. The OER proceeds via a four-electron process and involves the formation of two oxygen–oxygen bonds, and thus determines the overall rate kinetics of the water electrolysis [5]. Theoretically, OER requires a thermodynamic potential of 1.23 V (25 °C/1 atm); however, the commercial catalysts require much higher overpotentials (η). To date, Ir- and Ru-based electrocatalysts have been the best-performing OER electrocatalysts in aqueous media. However, their scarcity and prohibitive cost pose a major obstacle towards widespread use of electrolysis technologies. Therefore, significant efforts have been made in recent years to develop efficient and cost-effective OER catalysts that can enhance the performance of water electrolysis and enable the large-scale production of green hydrogen [6,7,8]. Of note, several 3D transition metal (TM)-based layered oxide electrocatalysts requiring lower η and demonstrating improved electrochemical performance toward OER have been reported recently [9,10].

Recently, larger interest has been shown in first-row TM-oxides (for instance, Cu, Co, Ni, and Fe) due to their enhanced electrochemical water splitting [11]. In particular, there has been significant interest in Cu-based oxidation electrocatalysts due to their natural abundance, low cost, rich redox chemistries, and non-toxic nature. Additionally, due to their low bandgap, they have been employed in several applications spanning from semiconductor, gas sensing, solar cells, and several biomedical applications [12,13,14]. However, their η values are still higher between 320–450 mV, which needs to be reduced significantly [15]. Another promising candidate is oxide of Mn (MnO_2_), which displays enhanced electrochemical performances due to its unsaturated edges and possesses a wide variety of electronic structures, and its natural abundance is an added advantage [15]. This material has been widely used in other energy domains such as energy storage and electrochemical sensors for detecting metal ions [16,17,18]. Inheriting the benefit of CuO and MnO_2_ by combining (CuO/MnO_2_), which is prepared hydrothermally, has been employed in catalytic applications for CO oxidations [19]. It is reported that CuO-MnO_2_ interfaces serve as the active sites for CO oxidations and the chemisorbed CO on CuO reacts with the oxygen species in MnO_2_. This combination has also been reported to enhance electrochemical performances during the oxidation of quinolone at supercritical conditions and the proposed model dictates that the catalytic reaction strongly follows the temperature dependency [20,21,22]. 

Interfacial systems of similar structural and chemical complexity and their structural, chemical, and adsorption properties, including in relation to real-world applications, have been studied by using high-level DFT and ab initio molecular dynamics calculations [23]. Molecular modeling is regularly used for guiding and controlling the synthesis of various materials [24]. The nanosheet arrays of metal oxide/carbon (MOx/C; M = Fe, Ag, and Mn) fabricated and employed in the covered two-dimensional (2D) metal–organic frameworks (2D-MOFs) exhibiting significant electrocatalytic activity and durability [25]. Nb-doped TiO_2_ rod-like particles were synthesized using molten salt as a reaction medium and applied as a catalyst support in ORR/OER bifunctional gas diffusion electrodes for use in metal–air batteries [26]. Ultrathin amorphous MnO_2_ modified prawn-shell-derived porous carbon (U–MnO_2_/PSNC) was designed and synthesized via a self-template-assisted pyrolysis coupling in situ redox reaction strategy to create robust oxygen electrocatalysts [27]. Fe_2_O_3_ decorated on carbon nanotubes as a promising architecture was utilized in OER reactions [28]. Pt alloy integrated in a cobalt–nitrogen–nanocarbon matrix 11.7 times higher than that of a commercial Pt catalyst retained a stability of 98.7% after 30,000 potential cycles [29]. A carbon/rGO composite, as an efficient palladium electrocatalyst for formic acid oxidation reaction, was reported [30]. Developments in electrocatalyst corrosion chemistry, including corrosion mechanisms, mitigation strategies, and corrosion syntheses/reconstructions based on typical materials and important electrocatalytic reactions, have been extensively studied [31]. These interesting reports encouraged us to produce a CuO/MnO_2_ system for electrocatalytic studies.

In the present study, we proposed a microwave-assisted synthetic protocol for the successful preparation of CuO and its interlaced architecture with MnO_2_. The structural reliability and morphological properties of the composites were confirmed with the aid of XRD, FESEM, XPS, and EDS mapping. The surface property of CuO/MnO_2_ was determined by BET analysis. As a proof-of-concept demonstration, we explored the virgin CuO and its composite CuO/MnO_2_ for application in the oxygen evolution reaction. The CuO/MnO_2_ exhibited an OER of 294 mV, which was evaluated with virgin CuO electrocatalyst (316 mV) at 10 mA cm^−2^ and remained stable for 15 h. The synthetic procedure and the OER property of the prepared materials demonstrate its potential applications in electrocatalysis for the design of new hybrid materials. 

## 2. Experimental Section

### 2.1. Materials and Reagents

The following materials of the highest purity were acquired from Sigma-Aldrich chemical (Korea) and applied without any further cleansing: CuSO_4_·5H_2_O, KMnO_4_, Mn(CH_3_COO)_2_·4H_2_O, and NaOH. Double-deionized water (DDW) was used for the preparation.

### 2.2. Syntheses of CuO

The protocol reported in [22] was utilized for preparing the CuO. Here, 20 mL of 0.2 M CuSO_4_·5H_2_O (20 mL) and 0.8 M NaOH (20 mL) solution were combined in a crucible and subjected to microwave irradiation for 10 min at 300 W. After completion of the reaction, the crude product was filtered, sequentially cleansed with deionized water to remove the water-soluble impurities, and ultimately dried at 80 °C. The obtained CuO was then utilized for electrocatalytic studies.

### 2.3. Syntheses of CuO/MnO_2_

First, 0.2 M Mn(CH_3_COO)_2_·4H_2_O (20 mL), 0.2 M KMnO_4_ (20 mL), and 100 mg of CuO were placed in a crucible. The reaction mass was subjected to microwave irradiation for 15 min at 300 W. The solid product was then filtered, washed, and dried at 80 °C. Calcination of the resultant product for 6 h at 400 °C at a heating level of 5 °C/min resulted in pure CuO/MnO_2_, which was then utilized for electrocatalytic studies.

## 3. Results and Discussion

Pristine CuO and its composite CuO/MnO_2_ were prepared using a microwave-assisted synthetic protocol. The precursors CuSO_4_.5H_2_O, Mn(Ac)_2_·4H_2_O, and KMnO_4_ were used for the formation of CuO/MnO_2_ as illustrated in Figure 1. The suggested imitation route was ecologically friendly, as CuO is interlaced with MnO_2_. Growth mechanism for the formation of CuO/MnO_2_ structure, can be divided into three stages. Firstly, the MnO^4-^ nuclei are produced and adsorbed on surfaces of CuO, forming MnO_2_ nuclei. With the increase of reaction time, the MnO_2_ nuclei are aggregated and transformed to nanosheets and nanorods. The MnO_2_ nanoroads are compact and totally cover surfaces of CuO, resulting in the formation of the hierarchical CuO/MnO_2_ nanocomposites. Such a process is supported by the morphological evolution at different growth stages via tuning the reaction time. It is worth noting that the interconnected MnO_2_ nanosheets and CuO nanostructure give rise to a highly porous morphology, which can offer very high surface area and many active sites for electrochemical transportation. X-ray diffraction (XRD) was used to study the structural advancement and phase purity of the CuO and its composite CuO/MnO_2_. As illustrated in Figure 2a, the phase analysis was conducted using XRD at Bragg’s diffraction angle range was adjusted between 5°–65° during the measurement, and the scan rate was set to 2°/minute. The signals for planes (110), (002), (111), (202), (020), and (202) were, respectively observed at 32.4°, 35.6°, 38.7°, 48.9°, 53.3°, and 58.1° for the monoclinic CuO structure (JCPDS card no. 89-2529). However, after incorporating MnO_2_, the supplementary diffraction heights detected at 2θ estimates of 12.7°, 15.6°, 18.0°, 37.6°, 40.6°, 42.0°, 49.9°, and 60.2° were, respectively credited to the (110), (001), (200), (121), (111) (301), (411), and (521) planes of the MnO_2_ component (JCPDS card no. 72-1982 and 80-1098), which is crystalline [32,33]. The surface areas (BET) and pore size distribution (BJH) parameters were studied using N_2_ adsorption−desorption isotherms as displayed in Figure 2, which demonstrated a characteristic IV isotherm that was ascribed to the mesoporous architecture of the aggregated CuO sphere, as displayed in Figure 2b. The CuO/MnO_2_ revealed an enhanced volume and diameter of the pores, as shown in Figure 2c. The specific surface area for CuO/MnO_2_ was found to be 54.50 m^2^ g^–1^, compared to 19.8 m^2^ g^–1^ for pure CuO. The enhancement in surface area of the CuO/MnO_2_ was a result of the combined porous MnO_2_ nanorods. 

The valency and chemical binding on the surface of the combined material were studied using X-ray photoelectron spectroscopy (XPS) analyses. The XPS spectra of CuO/MnO_2_ unveiled the manifestation of three elements (copper at 932.6 eV, manganese at 641.6 eV, and oxygen at 532.9 eV) on the exterior of the fused material. The binding powers of 932.6 and 952.4 eV referenced Cu 2p_3/2_ and Cu 2p_1/2_, and a spin–orbit separation of 19.8 eV was exhibited for Cu 2p, as displayed in Figure 2d. Furthermore, two peaks were detected at 942.7 and 960.9 eV, which further confirmed that the sample contained CuO. The Mn 2p demonstrated peaks at 652.9 and 641.6 eV with a partition of 11.3 eV, as shown in Figure 2e, which is in agreement with the literature [34,35,36]. The CuO/MnO_2_ presented a divided energy of 5.8 eV due to Mn−O in the existence of Cu^2+^. The O 1s exhibited three peaks (Figure 2f) at 532.4, 531.3, and 530.4 eV. The peaks of 532.4 and 531.3 were assigned to oxygen and moisture absorption on the composite surface. However, the peak at 530.4 eV agreed with the O^2-^ bonds with copper and manganese. Subsequently, the XPS results authorized the establishment of CuO/MnO_2_ composite, and it was endorsed by XRD results. The survey spectra for CuO/MnO_2_ are displayed in Appendix A.

The microstructure analysis of the CuO and CuO/MnO_2_ was determined by scanning electron microscopy (SEM) and transmission electron microscopy (TEM), and the findings were reviewed in Figure 3 and Figure 4. As illustrated in Figure 3a,b, the SEM images of CuO revealed the formation of amalgamated nanostructures. The energy-dispersive spectrometry (EDS) mapping directed the presence of O and Cu, as shown in Figure 3c,d.

As shown in Figure 4a,b, CuO/MnO_2_ was comprised of regular roads with negligible sheets at a 500 nanometer scale with accumulation. Moreover, at a greater enlargement, as shown in Figure 4b, the CuO/MnO_2_ structures demonstrated a plane surface with clearly visible nanorods. Furthermore, the images of CuO@MnO_2_ also confirm that MnO_2_ was clearly interlaced with CuO. The elemental arrangement of the CuO/MnO_2_ nanorod is exemplified in Figure 4c–f by energy-dispersive spectrometry (EDS) mapping, which indicated the presence of O, Cu, and Mn validating the effective construction of the composite. 

The LSV behavior of the CuO and CuO/MnO_2_ electrodes for OER was determined at a scan rate of 5 mV/s. Figure 5a displays the LSV curves of the CuO and CuO/MnO_2_ electrodes. The overpotentials of the CuO and CuO/MnO_2_ electrodes were 316 and 294 mV at a current density of 10 mA/cm^2^, respectively (Appendix A). Moreover, the CuO/MnO_2_ electrode exhibited a lower overpotential than that of the CuO electrode because of the enhanced intrinsic reaction kinetics and high catalytically active sites of the composite system of CuO/MnO_2_. Furthermore, the overpotential of the prepared catalysts was closely related to the Tafel slope values. Figure 5b displays the Tafel curves of the prepared CuO and CuO/MnO_2_ electrodes. The Tafel values of the CuO and CuO/MnO_2_ electrodes were determined from the following equation: η = b log (J) + a(1)
where b, a, and η are the Tafel slope, arbitrary fitting parameter, and overpotential, respectively. The Tafel values of the CuO and CuO/MnO_2_ electrodes were determined as 80 and 77 mV/dec, respectively. Compared to the CuO electrode, the CuO/MnO_2_ electrode exhibited a low overpotential and a small Tafel value, due to its large number of catalytic active sites and the improved intrinsic reaction kinetics. The noted excellent electrocatalytic behavior had an impact on the electrochemical charge transfer kinetics, which can be seen through electrochemical impedance spectroscopy (EIS) measurement. Figure 5c presents the Nyquist plots of the CuO and CuO/MnO_2_ electrodes, which were performed in a frequency range of 1 Hz to 10 kHz. The obtained semicircle at intermediate frequency range (1 Hz to 10 KHz) was attributed to the interfacial resistance within the grain itself. It affirmed that CuO/MnO_2_ was exhibiting a low series resistance (Rs) of 1.85 Ohms and low charge transport resistance (R_ct_) of 14.38 Ohms than the pure CuO-based electrode (Rs = 1.92 Ohm; R_ct_ = 21.77 Ohm). The EIS data are summarized in Appendix A, revealing that the electron transport in the material interior, particularly at the CuO and MnO_2_ interface, was increased compared to utilizing pure CuO, causing positively advanced electrocatalytically active sites that helped to improve the electrochemical reaction kinetics and yield better catalytic OER activity. Long-term potential stability is a crucial factor in catalysts for industrial purposes. Figure 5d presents the stability curve of the CuO/MnO_2_ electrode at 10 mA/cm^2^. The CuO/MnO_2_ electrode clearly exhibited a stable static voltage curve at 15 h, indicating the excellent stable nature of the prepared materials. The Over potential, Tafel slope of CuO and CuO/MnO_2_ is shown in Appendix A. Furthermore, the SEM images and XRD after OER confirms that the structure is retained (Appendix A). The embroidering layer of CuO/MnO_2_ proposes conduction channels which speed up electron transport and fundamentally enrich the electrochemical properties [37,38,39,40,41,42,43]. A comparative table for the electrochemical OER performances of copper-based materials is shown in Appendix A. The LSV polarization curve for IrO_2_ catalyst is shown in Appendix A.

## 4. Summary and Conclusions

In summary, CuO and CuO/MnO_2_ nano catalysts were synthesized using a green microwave-assisted protocol for utilization in OER. The prepared nano catalyst demonstrated the interlacing of CuO with MnO_2_, required a minimal η of 294 mV to achieve a benchmark current density (j = 10 mA cm^−2^), and a superior stability of more than 15 h. These enhanced electrochemical performances are attributed to the large volume and expanded diameter of the pores, which offer ample surface area for catalytic reactions to boost OER. Furthermore, the rate kinetics of the OER were favored in composites due to a low Tafel slope (77 mV/dec) compared to CuO (80 mV/dec). The improved OER and long duration stability, along with the easy and scalable fabrication process of the CuO/MnO_2_, offer a promising approach for the potential use of the electrode as an inexpensive catalyst material for electrochemical water-splitting applications.

## Figures and Tables

**Figure 1 nanomaterials-13-02329-f001:**
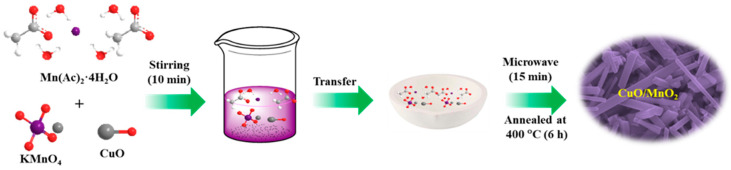
Schematics for synthesis of nanocomposite CuO/MnO_2_.

**Figure 2 nanomaterials-13-02329-f002:**
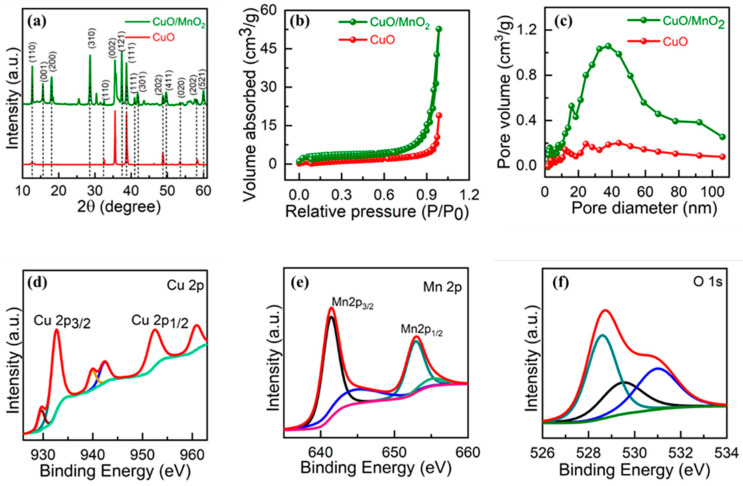
(**a**) XRD (**b**,**c**), BET data, and (**d**–**f**) XPS analysis of CuO/MnO2 (**a**) Cu 2p, (**b**) O 1s, and (**c**) C 1s.

**Figure 3 nanomaterials-13-02329-f003:**
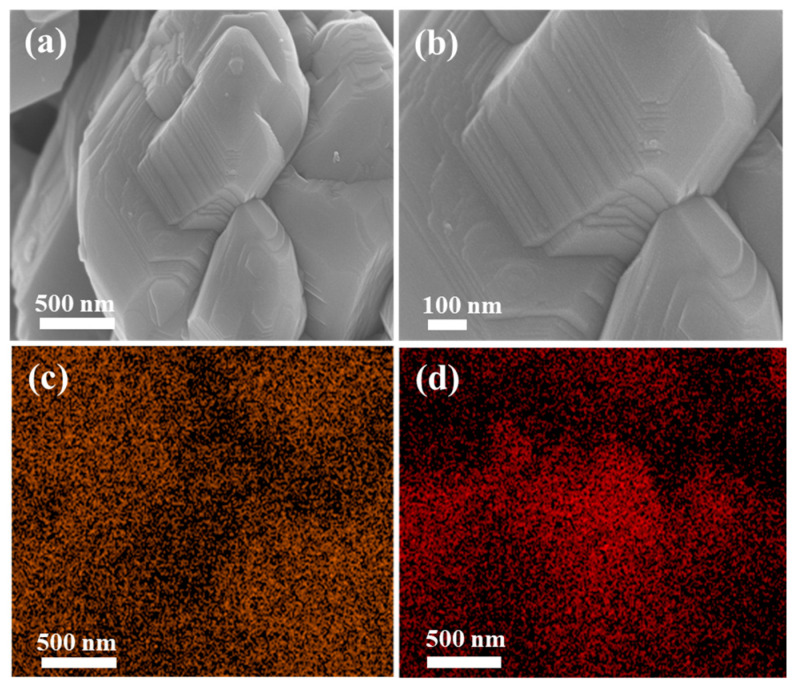
(**a**,**b**) SEM images; (**c**,**d**) EDX for CuO.

**Figure 4 nanomaterials-13-02329-f004:**
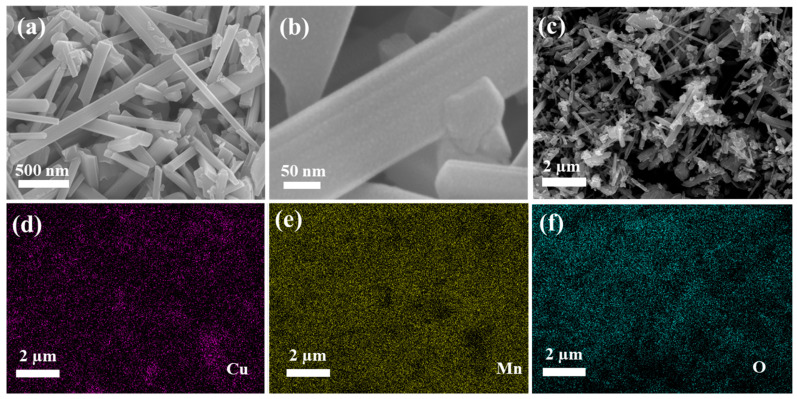
(**a**,**b**) SEM images; (**c**–**f**) EDX of CuO/MnO_2_.

**Figure 5 nanomaterials-13-02329-f005:**
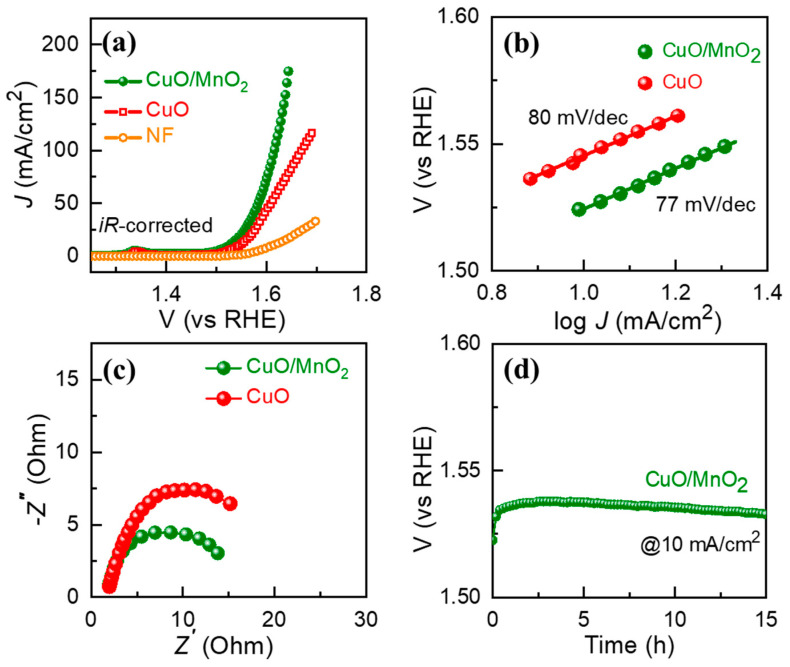
(**a**) iR-corrected LSV polarization curves, (**b**) Tafel plots, (**c**) Nyquist plots, and (**d**) long-term stability of the CuO/MnO_2_ catalyst.

## Data Availability

All the experimental data presented within this article along with the supplementary information will be made available from the authors upon reasonable request.

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
