# Peer review of "Self-Assembly of Copper Oxide Interfaced MnO2 for Oxygen Evolution Reaction"

_nanomaterials, 2023, doi:10.3390/nano13162329_

Round 1

Reviewer 1 Report

The manuscript entitled "Self-assembly of copper oxide interfaced MnO2 for Oxygen evolution reaction" discusses the self-assembly of electrocatalysts for the oxygen evolution reaction (OER), a process of great interest for its potential applications in renewable energy systems, such as water splitting for hydrogen production. While the paper provides valuable insights into the self-assembly of OER catalysts, it also exhibits several flaws that should be addressed before considering for publication. Following are some concerns.

1.       Authors could not discuss the novelty of the prepared self-assembly approach; a comparative analysis with other state-of-the-art OER catalysts is necessary to validate the catalyst's novelty.

2.       Authors are suggested to highlight the importance of the work in 1-2 sentences and then elaborate on the present findings concisely without citing detailed numerical data. Abstract should be simple and concise, avoid vague sentences and any unnecessary adjectives; for example, engineering efficient catalytic systems for electrochemical studies is a challenging area of investigation' exceptional morphology and brilliant electrochemical performance—extremely efficient and sensible electrocatalyst. Furthermore, authors should clearly state the potential applications and impact of their work in renewable energy systems.

3.       Ir-based catalysts are the state-of-the-art OER catalyst, authors are suggested to provide a thorough characterization, activity and stability comparison of the synthesized catalyst with Ir-based catalysts to signify the practical applications of OER catalysts.

4.       The long-term stability of the copper oxide-MnO2 interface remains a concern. During repeated cycles of OER, the catalyst may undergo structural changes, leading to the degradation of the interface and subsequently compromising catalytic performance. Authors have mentioned cyclic stability in keywords but didn't present any cyclic stability, except the stability at a constant potential. Authors are suggested to provide post-durability morphology analysis, especially after potential cycling for at least 10k cycles.

5.       The manuscript lacks a detailed discussion, especially since it is hard to find a discussion related to claims of interesting findings about this work. Authors should discuss results with reference to literature rather than just presenting results. Also, the abstract and conclusion shouldn't be overlapped. Furthermore, the abstract and conclusion should be distinct, and the conclusions should be supported by the data and discussion presented in the paper.

6.       Manuscript should be enriched with recent literature references, recommended references are 10.1038/s41467-022-34444-w, 10.1002/celc.201700879, 10.1002/adma.202200840

Moderate editing of English language required

Reviewer 2 Report

The authors systematically investigate the self-assembly of copper oxide interfaced MnO2 for oxygen evolution reaction. This is an original and timely task which was addressed by a well-chosen synthesis steps of CuO and the CuO/MnO2, presented in all the necessary detail. In addition, the characterization studies by XRD, XPS, SEM and EDX are both well-conducted and provide consistent and insightful results helping all interested readers understand the chemical and structural features of the samples. Characterization choices are motivated and explained, especially having in mind the complexity of the materials system studied. Thus, the results look credible besides being very well presented. Equally important, the discussion is feasible and fits the current research questions regarding multiple applications.

Summarizing, this work represents a valuable contribution with possible wider impact in the field of transition metal oxides.

The authors chose an adequate structure of the manuscript. Concise, and nicely illustrated figures and their corresponding analysis are provided.

There are some minor issues with this already very good manuscript that will need to be addressed before the manuscript becoming suitable for publication, i.e., it can be considered for publication after a minor revision:

1: Abstract should briefly but explicitly mention the main characterization techniques as adopted for the present study so it will be clear for the readers from the very beginning of the manuscript that adequate and sophisticated characterization was worked out.

2: The hydrothermal method is the method of choice for preparation of samples of the studied composition/interfacing and this is, experimentally, well investigated. In this relation, the thermal stability (upper thermal limit of stability) of the interfacial system should be briefly commented in order to support the credibility of present development work for the benefit of real-world applications in the future.

3: In the introduction, the authors miss that interfacial systems of similar structural and chemical complexity and their structural, chemical and adsorption properties including in relation to real-world applications have already been studied (and can be approached in the future) by using high-level DFT and ab initio molecular dynamics calculations, namely [Physical Chemistry Chemical Physics 25 (2023) 829-837; CrystEngComm 23 (2021) 6661-6667]. Such works need to be brought to context since modeling is frequently used for guiding and controlling their synthesis processes.

4: Spell-check and stylistic revision of the paper are necessary. Some long sentences, as well as misspellings, etc., are noticeable throughout the text.

Spell-check and stylistic revision of the paper are necessary. Some long sentences, as well as misspellings, etc., are noticeable throughout the text.

Round 2

Reviewer 1 Report

Authors have put their efforst to address the concerns by reviewer, I believe the manuscript improved with additional discussions. Therefore, I recommend the publication.

Authors are suggested to revise the manuscript carefully to omit typos and grammar errors.